# Image-Based Learning Using Gradient Class Activation Maps for Enhanced Physiological Interpretability of Motor Imagery Skills

**Diego F. Collazos-Huertas** *,† , **Andrés M. Álvarez-Meza** † and **German Castellanos-Dominguez** †

Signal Processing and Recognition Group, Universidad Nacional de Colombia, Manizales 170001, Colombia; amalvarezme@unal.edu.co (A.M.Á.-M.); cgcastellanosd@unal.edu.co (G.C.-D.)
* Correspondence: dfcollazosh@unal.edu.co
† These authors contributed equally to this work.

**Abstract:** Brain activity stimulated by the motor imagery paradigm (MI) is measured by Electroencephalography (EEG), which has several advantages to be implemented with the widely used Brain–Computer Interfaces (BCIs) technology. However, the substantial inter/intra variability of recorded data significantly influences individual skills on the achieved performance. This study explores the ability to distinguish between MI tasks and the interpretability of the brain's ability to produce elicited mental responses with improved accuracy. We develop a Deep and Wide Convolutional Neuronal Network fed by a set of topoplots extracted from the multichannel EEG data. Further, we perform a visualization technique based on gradient-based class activation maps (namely, Grad-Cam++) at different intervals along the MI paradigm timeline to account for intra-subject variability in neural responses over time. We also cluster the dynamic spatial representation of the extracted maps across the subject set to come to a deeper understanding of MI-BCI coordination skills. According to the results obtained from the evaluated GigaScience Database of motor-evoked potentials, the developed approach enhances the physiological explanation of motor imagery in aspects such as neural synchronization between rhythms, brain lateralization, and the ability to predict the MI onset responses and their evolution during training sessions.

**Keywords:** image-based learning; convolutional neural networks; motor imagery; GradCAM++; Brain–Computer Interface (BCI)

## 1. Introduction

The act of rehearsing body movements by means of their mental representation is known as motor imagery (MI). Brain–Computer Interfaces (BCIs) implement this paradigm by capturing brain activity patterns associated with elicited mental tasks and converting them into commands for external devices, having potential in a wide range of clinical, commercial, and personal applications [1,2]. Usually, the brain activity stimulated by MI is measured through Electroencephalography (EEG), which has several advantages: non-invasive data capture from the scalp surface, high temporal resolution, and inexpensive and portable devices with ease of setup [3]. Despite this, EEG data has a limited spatial resolution, and their weak electrical potentials make them susceptible to various disturbances, including electromyographic signals from the movement of the heart, eyes, tongue, and muscles, during data acquisition [4]. Furthermore, there is substantial variation in brain neural activity from one session to another and between different subjects [5]. This situation is partly explained because the EEG data used to drive effectors in MI-related BCI systems are profoundly affected by the ongoing mental states. Hence, the individual ability to generate mental imagery signals with a higher SNR of movement dictates how easily the neural responses elicited will be detectable [6]. In other words, a set of neurophysiological and non-neurophysiological causes in the brain structure or function of evaluated

subjects might be different from each other, making the implemented BCIs unfit for use on a few individuals [7,8]. As a result, a critical factor for the widespread application of these EEG-based systems is the significant performance differentiation in MI coordination skills among individuals. Therefore, using conventional classification algorithms may result in many subjects having poor accuracy, less than 70% (poor MI coordination skills), meaning that between 15–30% of the population struggle to improve their ability to monitor BCI systems concerning mental imagery tasks [9].

There are several strategies for improving BCI performance to enhance MI-BCI skills, such as using subject-specific designs and subject-independent set-ups. The former is designed and trained per subject, requiring prior calibration and rigorous system adaptation, and resulting in time-consuming and inconvenient BCI systems [10]. The latter instead trains a generalized model to be used by newly incorporated subjects. To this end, most BCI systems rely on temporal, spectral, and spatial features mainly computed on a single-trial basis to distinguish different MI patterns. In particular, since MI is a task-related power modulation of brain rhythms mainly localized in the sensorimotor area, Filter Bank Common Spatial Pattern (FBCSP) is a widely employed algorithm for effectively extracting EEG features by introducing an array of spatial bandpass filters [11]. However, these handcrafted feature extraction methods have the disadvantage of low temporal resolution, poor generalization over multiple subject sets, and a complicated tuning process [12]. As a solution for ineffective discriminability in decoding MI tasks using EEGs, Deep Learning (DL) algorithms have increasingly been applied to boost the classification accuracy of subject-independent classifiers. Among DL models, convolutional neural networks (CNN) with kernels that share weights for multidimensional planes have achieved outstanding success in extracting, directly from raw EEG data, unlocked local/general patterns over different domain combinations like time, space, and frequency [13–18]. The earlier layers learn low-level features, while the deeper layers learn high-level representations. DL architectures can also be adapted to allow learning models to mimic extracting EEG features by imposing explicit properties on the representations learned [19]. However, a few issues remain challenging for applying CNN learners to achieve accurate and reliable single-trial detection of MI tasks: (i) The DL outputs require collecting substantial training data for avoiding overfitting inherent of small datasets so that the provided superior performance comes at the expense of much higher time and computational costs [20]; (ii) finding representations extracted from EEG data invariant to inter-and intra-subject differences [21,22]; and (iii) for dealing with non-stationary and corrupted by noise artifacts, EEG-based training frameworks involve complex, nonlinear transformations that generate many trainable DL parameters, which in turn requires a considerable number of examples to calibrate them [23], fitting the data with inconvenient understanding. Consequently, the use of weights learned by CNNs tends to be highly non-explainable [24]. Although DL allows higher accuracy values, the outputs are usually learned by big, complex neural models, fitting the data with inconvenient understanding and thus tending to be highly non-explainable. Thus, the value of neural activity interpretation becomes evident in purposes like a medical diagnosis, monitoring, and computer-aided learning [25].

Recently, to promote understanding of their internal functioning, explainable CNN models have been devised with multiple 2D filtering masks that give insight into the role and derived contribution from intermediate feature layers, making the classifier's operation more intuitive [26,27]. Since the way the convolution kernels learn features within CNN frameworks directly influences final performance outcomes, visualizing the inputs that mainly excite the individual activation patterns of weights learned at any layer of the model can aid interpretation [28]. Several approaches to analyzing EEG decoding models via post hoc interpretation techniques are reported to enhance the ability to provide explainable information about sensor-locked activity across multiple BCI systems of diverse nature, introducing techniques like kernel visualizations, saliency map, EEG decoding components, score-weighted maps, ablation tests, among others [29–31]. Still, the approaches for building class activation mapping-based (CAM) visualizations have

increased interest in MI research, which performs a weighted sum of the feature maps of the last convolutional layer for each class using and a structural regularizer for preventing overfitting during training [32,33]. Specifically, the visualizations generated by gradient-based methods such as GradCam provide explanations with fine-grained details of the predicted class [34–36]. However, the CNN-learned features to be highlighted for interpretation purposes must be compatible with the neurophysiological principle of MI [37,38]. Despite the assumption that users possess mental skills that are developed to a certain degree, a lack of skills results in a mismatch between stimulus onset and elicited neural responses, incorrectly activating specific sensorimotor representations involved in the planning and executing motor acts [39]. One more aspect for visualization is the neural mechanisms of event-related Des/Synchronization. This contralateral activation, extracted from time and frequency bands over the sensorimotor cortex region, must be identified correctly to evaluate MI recognition's (in)efficiency [40,41]. Consequently, additional efforts are to be further conducted to improve the CNN-based training, aiming at better explaining spectral, temporal, and spatial behavioral patterns that act as constraints/guidelines in interpreting motor imagery skills [42,43].

This study explores the ability to distinguish between motor imagery tasks and the interpretability of the brain's ability to produce elicited mental responses with improved accuracy. As a continuation of the work in [44], we develop a Deep and Wide Convolutional Neuronal Network (D&W CNN) fed by a set of image-based representations (topoplots) extracted by the methods of wavelet transform and FBCSP from the multichannel EEG data within the brain rhythms. Further, we develop a visualization technique based on gradient-based activation maps (GradCam++), which enable the identification of multiple instances of the same class and multidimensional representations of inputs [45]. Each visualization map is extracted at different intervals along the MI paradigm timeline to account for intra-subject variability in neural responses over time. We also cluster the dynamic spatial representation of the extracted GradCam maps across the subject set using the Centered Kernel Alignment to come to a deeper understanding of MI-BCI illiteracy. According to the results obtained from the evaluated GigaScience Database of motor-evoked potentials, the developed approach enhances the physiological explanation of motor imagery in aspects such as neural synchronization between rhythms, brain lateralization, and the ability to predict the MI onset responses and their evolution during training sessions.

## 2. Materials and Methods

### 2.1. Deep and Wide CNN Learning from Image-Based Representations

We achieve a deep and wide learning architecture, which combines the benefits of memorization and generalization, through 2D topograms calculated for the multi-channel time-frequency features extracted by the CSP and CWT algorithms. The overall 2D feature mapping procedure is shown in Figure 1. To this end, we define $\{Y_n^z \in \mathbb{R}^{W \times H}, \lambda_n \in \Lambda\}$ as the input set holding $N$ labeled EEG-based topograms, where $Y_n^z$ is the $n$-th single-trial image with $H$ rows and $W$ columns extracted from every $z$-th set of image-based representations. Along with the topographic data, we also create the one-hot output vector $\lambda_n$ in $\Lambda \in \mathbb{N}$ labels. Of note, the triplet $z = \{r, \Delta_t, \Delta_f\}$ indexes a topogram estimated for each included domain principle $r \in R$ at the time-segment $\Delta t \in T$, and within the frequency-band $\Delta f \in F$.

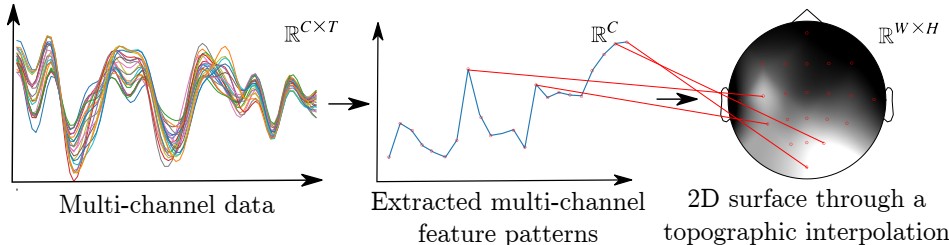

| Multi-channel data | Extracted multi-channel feature patterns | 2D surface through a topographic interpolation |

**Figure 1.** Performed 2D feature mapping procedure as EEG data preprocessing. Aiming at preserving the spatial interpretation, we transform the extracted multi-channel feature patterns onto a 2D surface through a topographic interpolation $\mathbb{R}^C \to \mathbb{R}^{W \times H}$ that maps each trial feature set as a two-dimensional circular view (looking down at the head top) using spherical splines.

Convolutional neural networks are DL architectures with a stacked structure capable of aggregating a bi-dimensional representation set $\{Y_n^z\}$, for which a set of included convolution kernels is shared to discover discriminating relationships, supporting a Multi-Layer Perceptron (MLP) to infer the one-hot output vector of labels $\lambda \in \Lambda$. Namely, a single probability vector $\tilde{\lambda} \in \Lambda$ is estimated by MLP at the last fully-connected layer $\psi_D$ with global average pooling, as follows:

$$\tilde{\boldsymbol{\lambda}} = \psi_D\{\cdot\} \circ \cdots \circ \psi_d\{\eta_d(\boldsymbol{A}_d\boldsymbol{u}_{d-1}+\boldsymbol{\alpha}_d)\} \circ \cdots \circ \psi_1\{\eta_1(\boldsymbol{A}_1\boldsymbol{u}_0+\boldsymbol{\alpha}_1)\}, \quad \{d = 0,\ldots,D\} \quad (1)$$

where $\boldsymbol{u}_0$ is the initial state vector that feeds the first fully-connected layer. Notation $\circ$ stands for function composition applied over each layer iteratively updated, $\eta_d:\mathbb{R}^{P'_d} \to \mathbb{R}^{P'_d}$ is the non-linear activation function of $d$-th fully-connected layer, $\boldsymbol{u}_d \in \mathbb{R}^{P'_d}$ is the hidden layer vector that encodes the convolved input 2D representations, $P'_d \in \mathbb{N}$ is the number of hidden units at $d$-th layer, the weighting matrix $\boldsymbol{A}_d \in \mathbb{R}^{P'_d \times P'_{d-1}}$ contains the connection weights between preceding neurons and the set $P'$ of hidden units at layer $d$, and $\boldsymbol{\alpha}_d \in \mathbb{R}^{P'_d}$ is the bias vector. Note that the hidden layer vector $\boldsymbol{u}_d \in \mathbb{R}^{P'_d}$ is iteratively updated at each layer, adjusting the initial state vector $\boldsymbol{u}_0$ to the flattened version of all concatenated matrix rows within the considered domains. That is, $\boldsymbol{u}_0 = \left[\text{vec}(\hat{Y}_L^z):\forall z \in Z\right]$ that sizes $W'H'Z \sum_{l \in L} I_l$ with $W' < W, H < H'$. Notation $[\cdots, \cdots,]$ stands for the concatenation operation.

The feature map at the last $L$ convolutional layer $\hat{Y}_L^z$ is the input of the MLP classifier and is generated by a stepwise 2D-convolutional operation performed over the input topogram set, as follows:

$$\hat{Y}_L^z = \varphi_L^z\{\cdot\} \circ \cdots \circ \varphi_l^z\{\gamma_l(K_{i,l}^z \otimes \hat{Y}_{l-1}^z + B_{i,l}^z)\} \circ \varphi_1^z\{\cdot\}, \quad \{l = 1,\ldots,L\} \quad (2)$$

where $\varphi_l^z\{\cdot\}$ is the $l$-th convolutional layer that holds the corresponding non-linear activation function $\gamma_l:\mathbb{R}^{W_l^z \times H_l^z} \to \mathbb{R}^{W_l^z \times H_l^z}$, $\hat{Y}_l^z \in \mathbb{R}^{W_l^z \times H_l^z}$ is the resulting 2D feature map of $l$-th layer, $\{K_{i,l}^z \in \mathbb{R}^{P \times P} : i \in I_l, z \in Z\}$ is the square-shaped layer kernel arrangement, $P$ is the kernel size and $i \in I_l$ is the number of kernels at $l$-th layer, and $B_{i,l}^z \in \mathbb{R}^{W_l^z \times H_l^z}$ is the bias matrix. The notation $\otimes$ stands for convolution operator.

Consequently, the predicted label probability vector $\tilde{\lambda}$ is computed within the framework that optimizes the CNN-parameter set $\Theta = \{K_{i,l}^z, \boldsymbol{A}_d, B_{i,l}^z, \boldsymbol{\alpha}_d\}$, as below:

$$\text{s.t.:} \quad \Theta^* = \arg \min_{K_{i,l}^z, \boldsymbol{A}_d, B_{i,l}^z, \boldsymbol{\alpha}_d} \{\mathcal{L}(\tilde{\boldsymbol{\lambda}}_n, \boldsymbol{\lambda}_n|\Theta); \forall n \in N\} \quad (3)$$

where $\mathcal{L}:\mathbb{R}^\Lambda \times \mathbb{R}^\Lambda \to \mathbb{R}$ is the gradient descent loss function $\mathcal{L}:\mathbb{R}^\Lambda \times \mathbb{R}^\Lambda \to \mathbb{R}$ employed to estimate the optimal values of weights $\{K_{i,l}^{z*}\}$, $\{A_d^*\}$ and bias $\{B_{i,l}^{z*}\}$, $\{\boldsymbol{\alpha}_d^*\}$. As widely used in deep learning methods, a mini-batch-based gradient is implemented by automatic differentiation and back-propagation procedures [46].

## 2.2. Gradient-Weighted Class Activation for Visualization of Discriminating Neural Responses

CAM is designed to visualize important regions of time-frequency representations of EEG segments and interpret predictions to improve the explanation of deep learning patterns. Thus, given a feature map set $\{\hat{Y}^i \in \mathbb{R}^{W \times H}\}$, a CAM representation $S \in \mathbb{R}^{W \times H}$ is computed by the linear combination $S = \sum_{\forall i} \Theta_i \cdot \hat{Y}^i$ that associates the CNN-classifier's ability to discriminate a particular class of $\Lambda$ with each spatial location of image-based representations. In order to account for the contribution from a complete stacked-layer set, the activation weights $\Theta_i$ are estimated by the following gradient-based differentiable learning algorithm of the activation maps (termed GradCam) [47]:

$$\Theta_i = \frac{1}{Q} \sum_{w,h} \partial \bar{\lambda} / \partial \hat{y}^i_{w,h} \tag{4}$$

where $Q$ is the number of activation map pixels, $\hat{y}^i_{w,h}$ holds the pixel of $i$-th feature map at $w, h$ position, and $\bar{\lambda}$ is the resulting classification score estimated for $\Lambda = \lambda$ that can be written as the linear combination, $\bar{\lambda} = \Theta^\top \bar{y}$, between the layer activation weights and the sum of all pixels across the feature maps, $\bar{y} \in \mathbb{R}^I$, having elements computed as $\bar{y}_i = \sum_{w,h} \hat{y}^i_{w,h}$ with $\bar{y}_i \in \bar{y}$. In practice, given an $i$-th feature map $\hat{Y}^i_l \in \mathbb{R}^{W^i_l \times H^i_l}$ together with its corresponding activation weight vector $\Theta_{i,l} \in \mathbb{R}^+$, the up-sampled GradCam version $S_l$ is estimated at each $l$-th convolutional layer, as follows [48]:

$$S_l = \mu\{\cdot\} \circ v_l \left( \sum_{i \in I_l} \Theta_{i,l} \hat{Y}^i_l \right) \tag{5}$$

where $v_l$ is a piece-wise linear function, and $\mu\{\cdot\}: \mathbb{R}^{W^i_l \times H^i_l} \to \mathbb{R}^{W \times H}$ is an up-sampling function. Then, the GradCam maps are fused via point-wise multiplication with the visualizations generated by Guided Back-propagation to obtain fine-grained pixel-scale representations [49].

For improving the object's localization ability, the learned estimates of $\Theta_i$ in Equation (4) are reformulated through a weighted averaged version across the pixel-wise gradients of each layer (GradCam++), as follows:

$$\Theta_{i,l} = \mathbf{1}^\top \left( G^i_l \odot v_l \left( \partial \bar{\lambda} / \partial \hat{Y}^i_l \right) \right) \mathbf{1}, \tag{6}$$

where notation $\odot$ stands for the Hadamard product, $\bar{\lambda}$ gathers a class-conditional score, $\mathbf{1}$ is an all-ones column vector of layer size, and matrix $G^i_l \in \mathbb{R}^{W^i_l \times H^i_l}$ is the weighting coefficient matrix for the gradients holding elements $g_{w,h} \in G^i_l$ approximated as below [45]:

$$g_{w,h} \simeq \sum_{\zeta \in H_l, \zeta' \in W_l} g_{\zeta\zeta'} \partial^3 \bar{\lambda} / g^3_{w,h} / \partial^2 \bar{\lambda} / g^2_{w,h} \tag{7}$$

Of note, the weighted combination in Equation (6) promotes dealing with different object orientations. At the same time, the non-linear-based thresholds in Equations (6) and (7) force to consider the contribution of just semi-positive definite gradients to $S_l$. Thus, $\Theta_i$ captures the importance of a particular activation map, and we prefer positive gradients to indicate visual features that increase the output neuron's activation, meaning that only the obtained visualization accounts for the time-frequency features with increasing output neuron's activation [32].

## 2.3. Clustering of Common GradCam Maps across Subjects Using Centered Kernel Alignment

We used group-level analysis to address MI skills enhancement by clustering the neural responses derived from the relevant spatial activations that could be considered characteristic and distinctive of a particular subject subset. As a result, to find common discriminatory

brain activations between subjects, we use the centered kernel alignment (CKA) approach (see [50]) that quantifies the similarity between kernelized spaces based on the projection of the GradCam++ set onto a reproducing kernel Hilbert space. Optimization of spatial map relevance implies adjusting the GradCam++ inputs, $\boldsymbol{\xi}_m = \left[\operatorname{vec}(\mathbb{E}\{S^{\lambda}_{m,n} : \forall n \in N\}) : \forall \lambda \in \Lambda\right]$, and output accuracy values $y_m \in \mathbb{R}[0,1]$ for each $m$-th subject with ($m \in M$). Notation $\mathbb{E}\{\cdot\}$ stands for the expectation operator. Therefore, for the subject set, we determine the matrix $\Xi \in \mathbb{R}^{M \times J}$, holding each row vector $\boldsymbol{\xi}_m \in \mathbb{R}^J$ with $J = WHZ$, and the score vector $\boldsymbol{y} \in \mathbb{R}^M$.

As part of the CKA-based approach, we choose two kernels ($\kappa_{\Xi} : \mathbb{R}^J \times \mathbb{R}^J \to \mathbb{R}$ and $\kappa_{\boldsymbol{y}} : \mathbb{N} \times \mathbb{N} \to [0,1]$) to assess both similarity matrices: one between GradCAM++ inputs, $\boldsymbol{V}_{\Xi} \in \mathbb{R}^{M \times M}$, and another between the accuracy outputs $\boldsymbol{V}_{\boldsymbol{y}} \in [0,1]^{M \times M}$, which are, respectively, described as:

$$\kappa_{\Xi}(\boldsymbol{\xi}_m, \boldsymbol{\xi}_{m'}|\boldsymbol{\Sigma}) = \exp\left(-\|\boldsymbol{\xi}_m\boldsymbol{\Sigma} - \boldsymbol{\xi}_{m'}\boldsymbol{\Sigma}\|_2^2/2\right), \tag{8}$$

$$\kappa_{\boldsymbol{y}}(y_m, y_{m'}) = \delta(y_m - y_{m'}), \tag{9}$$

where $\delta(\cdot)$ is the delta function, and $\boldsymbol{\Sigma} \in \mathbb{R}^{J \times J'}$ ($J' \leq J$) is a projection matrix.

As a next step, we calculate $\boldsymbol{\Sigma}$, intending to highlight relevant spatial feature combinations from $\Xi$ through the cost function within the optimizing CKA framework, as follows [51]:

$$\hat{\boldsymbol{\Sigma}} = \arg\max_{\boldsymbol{\Sigma}} \log \frac{\langle \bar{\boldsymbol{V}}_{\Xi}(\boldsymbol{\Sigma}), \bar{\boldsymbol{V}}_{\boldsymbol{y}}\rangle_{\mathrm{F}}}{\|\bar{\boldsymbol{V}}_{\Xi}\|_{\mathrm{F}} \|\bar{\boldsymbol{V}}_{\boldsymbol{y}}\|_{\mathrm{F}}}, \tag{10}$$

where $\boldsymbol{V}_{\Xi}(\boldsymbol{\Sigma})$ highlights the dependency of $\kappa_{\Xi}$ with respect to the projection matrix in Equation (8), $\hat{\boldsymbol{\Sigma}}$ is the centered kernel matrix computed as $\hat{\boldsymbol{\Sigma}} = \tilde{\boldsymbol{I}}\boldsymbol{\Sigma}\tilde{\boldsymbol{I}}$, being $\tilde{\boldsymbol{I}} = \boldsymbol{I} - \boldsymbol{1}_M^{\top}\boldsymbol{1}_M/M$ the centering matrix, $\boldsymbol{I} \in \mathbb{R}^{M \times M}$ is the identity matrix, $\boldsymbol{1}_M \in \mathbb{R}^M$ is the all-ones vector, and $\langle \cdot, \cdot \rangle_{\mathrm{F}}$ and $\|\cdot\|_{\mathrm{F}}$ stand for the Frobenius inner product and norm, respectively.

The mapping matrix $\boldsymbol{\Sigma}$ is computed by the gradient-based approach estimated for Equation (10), yielding:

$$\nabla_{\boldsymbol{\Sigma}}\rho\big(\boldsymbol{V}_{\Xi}(\boldsymbol{\Sigma}), \boldsymbol{V}_{\boldsymbol{y}}\big) = -4\Xi^{\top}\big(\nabla_{\boldsymbol{V}_{\Xi}}\rho\big(\boldsymbol{V}_{\Xi}(\boldsymbol{\Sigma}), \boldsymbol{V}_{\boldsymbol{y}}\big) \odot \boldsymbol{V}_{\Xi}(\boldsymbol{\Sigma})\big) - \dots \tag{11}$$
$$\cdots - \operatorname{diag}\Big(\boldsymbol{1}^{\top}\big(\nabla_{\boldsymbol{V}_{\Xi}}\rho\big(\boldsymbol{V}_{\Xi}(\boldsymbol{\Sigma}), \boldsymbol{V}_{\boldsymbol{y}}\big) \odot \boldsymbol{V}_{\Xi}(\boldsymbol{\Sigma})\big)\Big)\Xi\boldsymbol{\Sigma}$$

where $\operatorname{diag}(\cdot)$ is the diagonal operator and $\odot$ is the Hadamard product.

Consequently, the cost function in Equation (10) measures the matching between the spatial representation achieved by GradCam++ maps $S$, coded by the projection matrix $\boldsymbol{\Sigma}$, and the classification accuracy space.

## 3. Experimental Set-Up

This paper examines the proposed approach for improving the post hoc interpretability of Deep and Wide convolutional neural networks based on the Class Activation Maps of GradCam++ to characterize the inter/intra-subject variability of the brain motor-elicited potentials. This purpose is addressed with an evaluation pipeline that includes the following stages (see Figure 2): (*i*) Preprocessing and feature extraction of the time-frequency representations from raw EEG data to resemble the FBCSP algorithm and the CWT method. Then, the extracted multi-channel features are converted onto a 2D surface through a topographic interpolation to feed the CNN: (*ii*) Bi-class discrimination of motor-evoked tasks within a D&W CNN framework and estimation of the resulting weighted feature representations through the generalization version of Class Activation Mapping. For simplicity, gray-scale images and a binary classification problem are considered; and (*iii*) Group-level relevance analysis is performed across the subject groups through a CKA projection of the relevant spatial activations obtained from the GradCam++ maps.

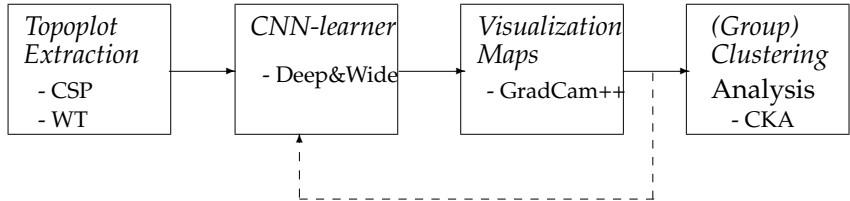

**Figure 2.** Guideline of proposed the spatial relevance D&W CNN framework using GradCam++ for enhanced physiological interpretability of motor imagery skills.

**Database of Motor Imagery Potentials**

*GigaScience* (publicly available at http://gigadb.org/dataset/100295 (accessed on 1 August 2021)): This collection holds EEG data obtained from fifty-two subjects (though only fifty are available) according to the BCI experimental paradigm of MI. As shown in Figure 3, the paradigm begins with a fixation cross presented on a black screen within 2 s. Then, a cue instruction appeared randomly on the screen during 3 s to ask each subject to imagine moving the fingers, starting to form the forefinger, and proceeding to the little finger touching each to their thumb. Then, a blank screen appeared at the start of the break, lasting randomly between 4.1 and 4.8 s. For either MI class, these procedures were repeated 20 times within a single testing run. Data were acquired by a 10-10 placement $C$-electrode system $C = 64$ with 512 Hz sampling rates, collecting 100 trials per individual (each one lasting $T = 7$ s) in two labeled tasks $\Lambda = 0$—left hand, or $\Lambda = 1$—right hand.

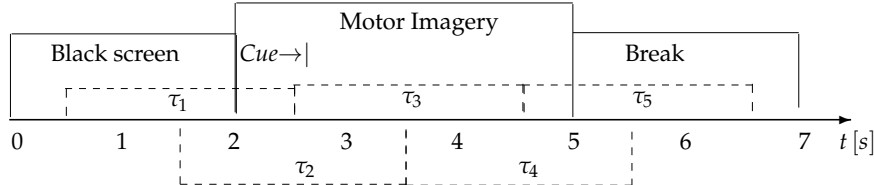

**Figure 3.** Timeline of the evaluated motor imagery paradigm.

**Preprocessing and Feature Extraction of Image-Based Representations**

At this stage, each raw channel is bandpass-filtered within [8–30] Hz using a five-order Butterworth filter. Further, as performed in [44], we carry out a bi-domain short-time feature extraction, namely, using two approaches: continuous wavelet transform and Filter Bank Common Spatial Patterns. In the former extraction method, CWT coefficients provide a compact representation of the EEG energy distribution, resulting in a time-frequency decomposition with components distinct from conventional Fourier frequencies. A feature set of CWT is extracted by using the Complex Morlet function commonly used in the spectral analysis of EEG, fixing the scaling value to 32. According to the second approach, a multi-channel EEG dataset can be mapped to a subspace with a lower dimension, or latent source space, to enlarge the class separation by maximizing its labeled covariance. Here, we set the amount of CSP components as $3\Lambda$ ($\Lambda \in \mathbb{N}$ is the number of MI tasks), utilizing a regularized sample covariance estimation.

For comparison purposes, the parameters of both extraction techniques are adjusted to the set, optimizing the accuracy performance of FBCSP [52]. That is, fixing the sliding short-time window length parameter $\tau = 2$ s with an overlapping step of 1 s, resulting in $N_\tau = 5$ EEG segments. For implementing the filter bank strategy, the following bandwidths of interest: $\Delta f \in \{\mu \in [8\text{--}12], \beta \in [12\text{--}30]\}$ Hz. These bandwidths belong to $\mu$, and $\beta$ rhythms, commonly associated with electrical brain activities provoked by MI tasks [53]. In order to generate a physiological interpretation according to the implemented experimental paradigm of MI, the dynamics are analyzed at the following representative intervals of interest: $\tau_1 = [0.5\text{--}2.5]$ s (interval prior to cue-onset or task-negative state), $\tau_2 = [1.5\text{--}3.5]$ s (cue-onset interval), $\tau_3 = [2.5\text{--}4.5]$ s (motor imagery interval), $\tau_4 = [3.5\text{--}5.5]$ s (decaying motor imagery interval), and $\tau_5 = [4.5\text{--}6.5]$ s (break period). Then, we use the resulting bandpass-

filtered and time-segmented EEG multi-channel data to calculate the 2D topographic maps building the $Y_n^z$ set, extracted to feed the proposed D&W CNN framework.

**Classification of MI Tasks Using Convolutional Neural-Networks**

The Deep and Wide architecture employed to support the brain neural discrimination is presented in Figure 4. The initial step of the MLP-based classifier is powered by the 2D maps extracted from the convolution network input topogram set. Setting of parameters is shown in Table 1 that has the following notations: $O = RN_\Delta N_\tau$, $N_\Delta$ denotes the number of filter banks, $P'$—the number of hidden units (neurons), $C$—the number of classes and $I_L$ stands for the amount of kernel filters at layer $L$.

To implement the optimizing procedure, we apply the Adam algorithm under the following fixed parameter values: learning rate of $1 \times 10^{-3}$, 200 training epochs, and a batch size of 256 samples. For assessment, mean square error (MSE) is selected as a loss function. That is, $\mathcal{L}(\tilde{\lambda}_n, \lambda_n | \Theta) = \mathbb{E}\{(\tilde{\lambda}_n - \lambda_n)^2\}$.

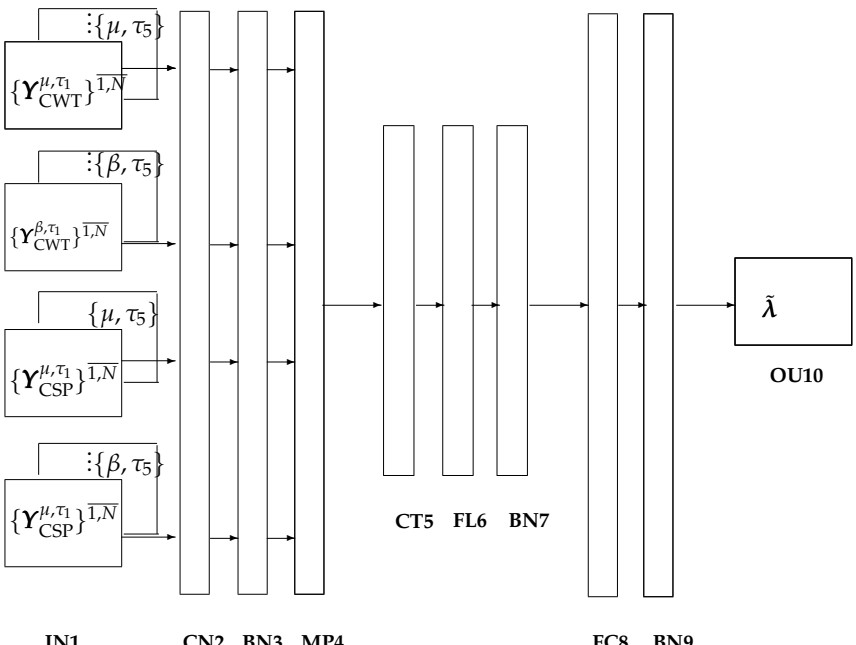

**Figure 4.** Scheme of the Deep and Wide architecture to support subject-oriented brain neural discrimination of labeled MI tasks using Convolutional Neural Networks.

**Table 1.** Detailed *Deep&Wide* architecture. Layer FC8 implements the regularization procedure using the *Elastic-Net* configuration, whereas layers FC8 and OU10 apply a kernel constraint adjusted to *max_norm(1.)*.

| Layer | Assignment | Output Dimension | Activation | Mode |
|---|---|---|---|---|
| **IN1** | Input | $[40 \times 40]$ | | |
| **CN2** | Convolution | $[40 \times 40 \times 2]$ | ReLu | *Padding* = SAME |
| | | | | *Size* = $3 \times 3$ |
| | | | | *Stride* = $1 \times 1$ |
| **BN3** | Batch-normalization | $[40 \times 40 \times 2]$ | | |
| **MP4** | Max-pooling | $[20 \times 20 \times 2]$ | | *Size* = $2 \times 2$ |
| | | | | *Stride* = $1 \times 1$ |
| **CT5** | Concatenation | $[20 \times 20 \times O \cdot I_L]$ | | |
| **FL6** | Flatten | $20 \cdot 20 \cdot O \cdot I_L$ | | |
| **BN7** | Batch-normalization | $20 \cdot 20 \cdot O \cdot I_L$ | | |
| **FC8** | Fully-connected | $[P' \times 1]$ | ReLu | Elastic-Net |
| | | | | *max_norm(1.)* |
| **BN9** | Batch-normalization | $[P' \times 1]$ | | |
| **OU10** | Output | $[C \times 1]$ | Softmax | *max_norm(1.)* |

## 4. Results

### 4.1. Achieved Accuracy of Implemented D&W CNN Classifier

We estimate the classifier performance evaluated through the cross-validation strategy for validation purposes, reserving the 90% of points as training, while the remaining group is a hold-out data set. To this end, we employ a stratified *K*-Folds cross-validation, adjusting *K*=5. Figure 5 displays the values of accuracy obtained by the Linear Discriminant Analysis algorithm fed by the features extracted using the FBCSP method (plotted as an orange line), as described in [52]. Based on this baseline discriminating method of handcrafted features, all subjects are ranked in decreasing order of reached mean accuracy, showing that more than half of the subjects fall under the 70%-level poor MI skills (indicated by the red line); this results in a very ineffective MI training scheme.

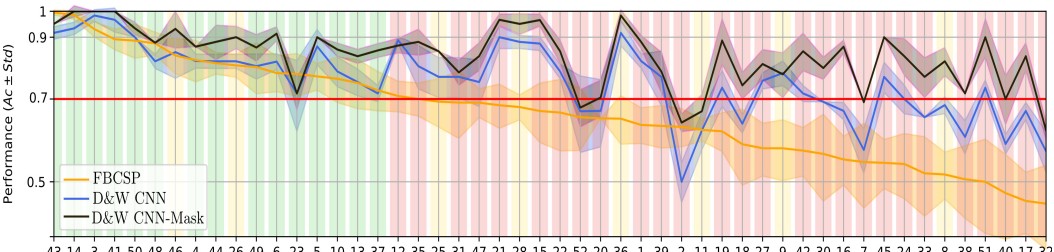

**Figure 5.** Achieved MI-related classification performance. Note: the red line at 70% level shows the poor MI coordination skill threshold, under the subjects are considered as worse-performing.

As a way of improving this analysis of the poor MI skills phenomenon, the subjects can be grouped according to their pre-training motor abilities that are critical to performing MI tasks. A common approach for clustering the individual set is to form partitions using both the features extracted from data of the state of wakefulness conscious together with the accuracy values achieved by one selected classifier under consideration [54]. Specifically, we compute the pre-training indicator that quantifies the potential for desynchronization over the sensorimotor area at the pre-cue interval evaluated in the 1-s window preceding the movement onset ($\tau_1$). Utilizing the accuracy values given by FBCSP, the Silhouette score-based cost yields three clusters to perform the *k*-means algorithm for estimating each subject membership of poor MI-BCI coordination skills group. Consequently, Figure 5 presents the obtained groups of motor skills painted in color bars: the best-performing subjects (Group I colored in green); the ones with intermediate MI abilities (Group II in

yellow); and Group III (red) is the set of individuals with the worst performance (half of the set, 27 individuals), as well as with an accuracy below 70%.

Next, as a strategy to reduce the problem of poor MI skills [55], we improved MI's accuracy by implementing a D&W CNN classifier (blue line), which increased it from $67.7408 \pm 5.89$ (FBCSP) to $76.218 \pm 3.96$ on average over the whole subject set. Nevertheless, the CNN-based accuracy can be further enhanced by modifying the set of input topograms through the activation maps estimated after the training stage, similarly as suggested in [56]. Namely, since the activation-based representation assesses the electrode contribution to the CNN-based accuracy, these maps can be employed to retrain the CNN-based classifier of each individual. The results of this post hoc analysis via these emphasizing masks (termed D&W CNN-Mask) are plotted in black to illustrate the further improvement in the classifier performance ($84.09 \pm 3.46$).

Therefore, GradCam maps improve the performance of worse-performing subjects, so the number of subjects under the poor MI skills level (i.e., having accuracy below 70%) decreases: D&W CNN yields 15 while D&W CNN-Mask—just six. The baseline FBCSP-based discriminating approach estimates a much higher amount of 32 poor-performing subjects, presenting the clustering of not compact groups, meaning that some subjects may be assigned to the wrong membership.

### 4.2. Grouping of Subjects with MI Skills Using Common GradCam++

Another strategy to deal with the poor MI coordination skills issue is to improve the clustering of individuals to explore common dynamic representations across the subject set. Applying the Mahalanobis distance, we achieve CKA matching that is thus fed by the activation maps and the corresponding D&W CNN accuracy vector. However, the alignment is followed by PCA dimensional reduction to deal with the enlarged dimensionality of obtained dynamic representation; each GradCam-based map is computed on the trial basis from both brain rhythms ($\mu$, $\beta$), at each time window $\tau$, using two feature extraction methods (CWT and FBCSP), and for either MI label $\boldsymbol{\Lambda}$. The result is a reduced feature space corresponding to the discriminating time-frequency representations (encoded in the attention maps) that contribute most to the classification performance.

In analogy to the partition above presented, we present the results of clustering the reduced feature space, for which the Silhouette score-based cost yields three clusters to feed the *k*-means algorithm. The scatter plot in Figure 6a depicts the obtained subject's membership of motor skills (painted in the same color bars as above), showing a good within-group separation. However, some outlier subjects affect the between-group compactness (namely, #11 and #12). According to the newly assessed partitions, Figure 6b displays the subjects ranked in decreasing order of the accuracy obtained by the D&W CNN classifier (blue line), showing that both outlier points scramble the corresponding groups of membership. It is worth noting that the newly estimated poor-performing subjects (G-III) are the only ones achieving accuracy scores under the 70%-threshold (red line), except for the outlier point #11.

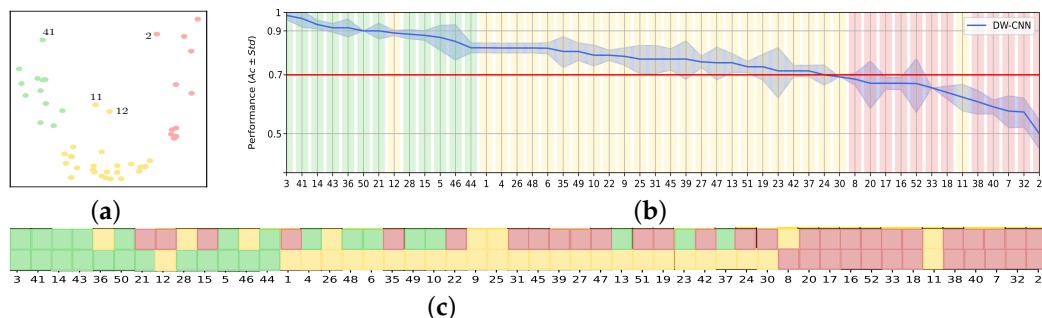

(a)                                                                                          (b)

(c)

**Figure 6.** Resulting clustering performed by the CKA-based data projection from GradCam++ activation maps. (**a**) Obtained subject's membership, (**b**) Ranking of the accuracy obtained by the D&W CNN classifier, and (**c**) Membership difference between the common GradCam representations and pre-training motor abilities.

For comparison, the arrangement in Figure 6c depicts the cells colored according to the individual clusters and shows the difference in membership assigned by the common GradCam representations (bottom row) and the above-explained pre-training motor abilities (top row). As seen, the former clustering is less scrambled and yields a more reduced number of subjects with the poor MI skills issue, resulting in the set of poor-performing individuals in Group III of the last partition. Notably, both outliers (#11 and #12) are apart from their designated group, regardless of the evaluated clustering approach.

Additionally, the feature maps computed are based on motor skills to determine the importance of each electrode location in terms of the CNN decision-making process. Thus, the interpretation of the spatial contributions improves in discriminating between both labels of intention recognition. Figure 7 displays the GradCam maps of tree subjects (each one belonging to a different group of MI skills) computed within the considered intervals of the MI paradigm timeline. As seen in the top row, the activation weights produced by subject # 41 (Group I) are low at the two initial windows (pre-cue and cue) because of a lack of stimulation. Instead, the maps reveal boosted neural activity during the following two intervals of MI responses (after cue). Moreover, we can observe a few clusters with powerful contributions that are rightly focalized over the sensorimotor area. Further, the contribution of neural brain activity decreases at the last window because of desynchronization before the break. Generally, the whole timeline of activations generated by this well-performing subject fulfills the MI paradigm.

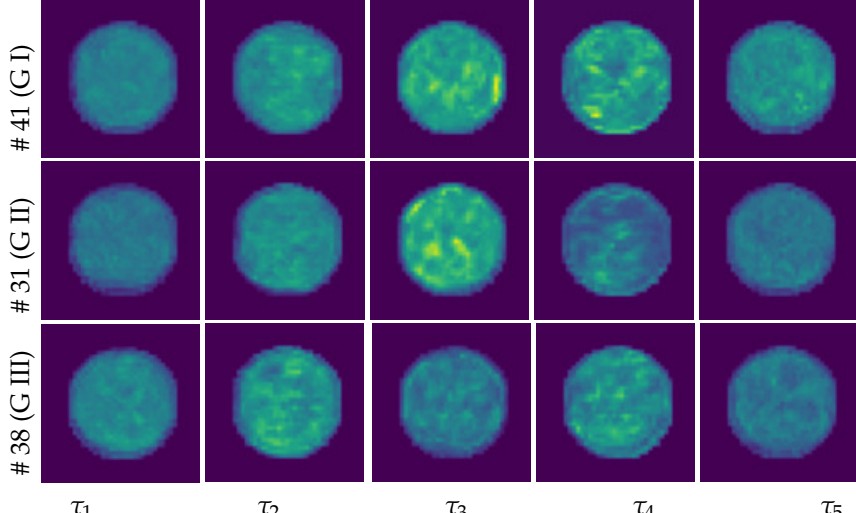

$\tau_1$          $\tau_2$          $\tau_3$          $\tau_4$          $\tau_5$

**Figure 7.** Max-pooling representation of GradCam++ maps achieved by the representative subjects: Top raw # 41 (G I), Middle: # 31 (G II), Bottom: # 38(G III).

A similar brain activity behavior holds to some extent for subject # 31 (Group II), but not entirely. In particular, the middle row presents the sequence of the corresponding Grad-Cam maps, revealing that the saliency of after-cue weights decreases over the neighboring interval $\tau_4$, suggesting that the D&W CNN classifier must extract discriminating information from shorter intervals of the elicited MI response. This reduction in contributing periods should be related to increased variability of responses, which becomes so high within $\tau_5$ that there are no relevant electrodes.

Lastly, the bottom row shows that the poor-performing subject (Group III) produces a GradCam map set with an even and weak contribution from each time window. However, the spatial relevance assessed for the cue interval also increases a little, being very active at the occipital and frontal electrodes, pointing out in some attentional interaction disorienting the individuals; this situation becomes untenable since the assessments of MI skills should be performed after the cue stimulation. Overall, the brain neural responses, shown for a worse-performing subject, barely satisfy the MI paradigm.

### 4.3. Averaged GradCam Maps over MI-Skills Groups

Aiming to explain the elicited brain activity according to the grouped motor skills, we compute the timeline sequence of activation maps averaged over each group of individuals and estimated apart for either label. The first aspect of visualization is the time-varying activations collected within the brain waveforms that are displayed in Figure 8. As seen, the $\mu$ band of the Group-I individuals contributes the most, except for the MI interval, $\tau_3$, meaning that the execution of elicited neural responses comprises the brain activity elicited at higher frequencies [57]. Although a similar behavior holds for Group-II individuals, the brain activity at higher $\beta$ frequencies is not so intense as for the well-performing subjects. In the case of Group-III individuals, the activation maps of $\mu$ are still noticeable, but have a significantly attenuated $\beta$-contribution at the interval, $\tau_3$. Poor-performing subjects are likely to have a low contribution from the upper spectral components as a result of the variability of the measured EEG data during the training and recording procedures. Several reasons may account for this finding. For example, the indications given by arrows or messages seem rather abstract, making imagining the corresponding movements challenging. As suggested in [58], practicing the conventional experiment paradigm frequently makes the subjects easily distracted and tired.

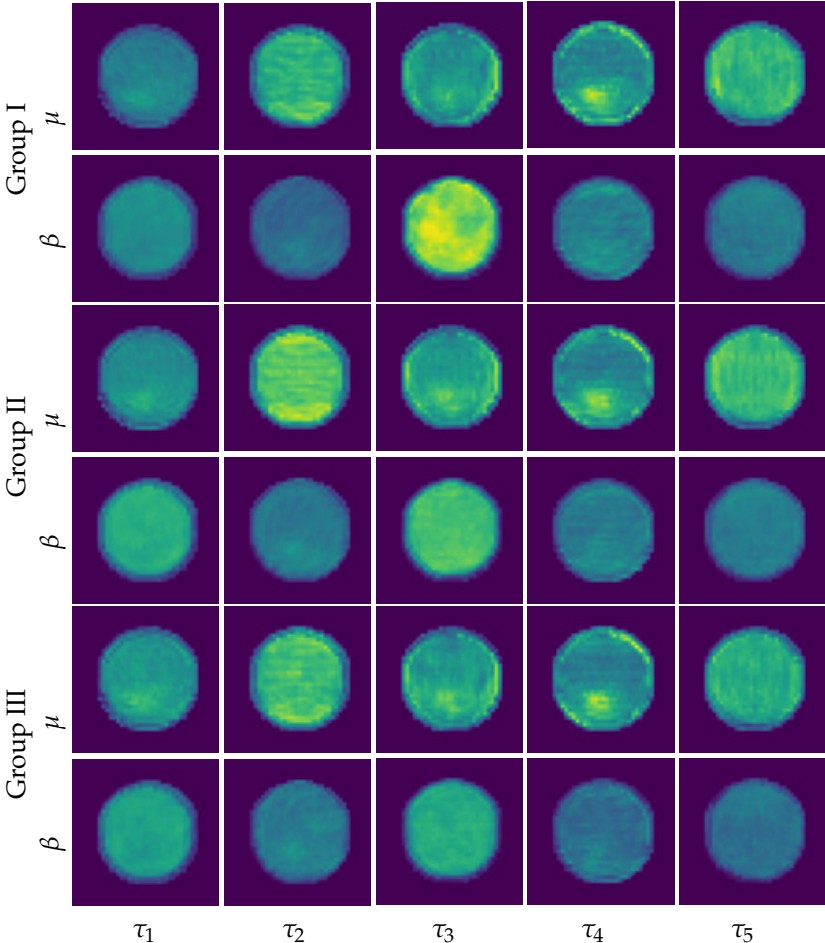

**Figure 8.** Timeline of GradCam++ maps achieved by the skills-ranked groups and collected within both rhythms: the lower spectral content, *μ*, and lower spectral content, *β*.

The next aspect of visualization is the brain lateralization of motor imagery related to the brain's anatomical structure and function that varies between left and right hemispheres [59]. In the figure below (see Figure 9a), the spatial contribution performed by Group I matches the label contribution of the subject with the best performance, as seen above in Figure 7. However, the label $\Lambda = 1$ holds an averaged activation map with more solid values, meaning that the corresponding neural activity provides a more discriminating contribution than the one from $\Lambda = 0$. Moreover, this behavior becomes more accentuated within the after-cue intervals of MI responses, with clearly focalized activity over the sensorimotor area. This asymmetric, increased contribution of $\Lambda = 1$ (right hand) can be related to the vast majority of tested subjects being right-handed dominant, and it is also observed in the averaged saliency of Group II. However, the discriminant spatial ability decreases at the after-cue interval for the label $\Lambda = 0$. Regarding the poor-performing subjects, the left-hand activation map of $\Lambda = 0$ is weak, while the right-hand imagery maps show an irregularly raised contribution before the MI intervals. This situation may be explained by the difficulty of subjects in practicing the MI paradigm.

Our approach resembles the physiological phenomenon of brain lateralization by splitting the training set into two labeled filter mask sets of GradCam maps to measure their contribution in identifying either hand class. As a result, we feed the D&W CNN classifier with the filtered input topogram set using the class activation map (as a filter mask) averaging across the trial set. Figure 9b displays the accuracy obtained, showing that the right-hand label (i.e., the left-hemisphere map set) marked in red overperforms the accuracy of the contralateral hemisphere (left-hand label) colored in blue (73.4 vs. 71.5,

respectively). Note the right-hand dominance of the best-performing subjects, which in contrast is not present in the subjects with the lowest accuracy.

### 4.4. Prediction Ability of Extracted GradCam++

One more strategy of dealing with the poor MI coordination skills is identifying the causes of inter and intra-subject variability and incorporating appropriate procedures to its compensation. Here, we assess the correlation between the neural activity features extracted in advance (pre-training indicator) with the MI onset responses and the evolution of learning abilities (training phase) [60].

*Pre-training indicator*: We estimate the pre-training prediction ability of the activation maps extracted from the pre-cue interval ($\tau_1$) for anticipating the subject's accuracy produced by the D&W CNN classifier in distinguishing either MI class. Thus, we obtain an *r*-squared value of $r2 = 0.36$ comparable with the one reported in [61] implemented via a similar D&W Neural-Network regression, implying that the activation maps may help in pre-screening participants for the ability to learn regulation of brain activity.

*Training phase indicator*: In evaluating the evolution of neural synchronization mechanisms over training sessions, we employ the inertia criterion as a measure of intra-class variability as used in [62]. Analogous to the within-cluster sum-of-squares criterion [63] that measure how internally coherent clusters are, inertia value $\epsilon$ indicates how coherent the estimated activation maps are: $\epsilon = \mathbb{E}\{(S_n - \hat{S})^2 : n \in N\}$, where $N$ is the number of activation maps within the subject's data set, and $\hat{S}$ is the averaged GradCam++ saliency map (i.e., $\hat{S} = \mathbb{E}\{S_n : \forall n \in N\}$). In the top row of Figure 10, the time-interval estimates of inertia values are plotted as they progress across the runs. As the Group-I subjects progress through the trials, they concentrate their discriminating neural responses over the onset intervals ($\tau_3$ and $\tau_4$), although the strength of the elicited brain activity drops in the last runs. Having fewer skills in Group II results in a spread of discriminating ability of the activation maps over the neighboring intervals. For the poor-performing subjects, this behavior becomes accentuated to the point that the extracted activation maps search for contributing neural activity even over time intervals before and during cue onset. Additionally, the bottom row in Figure 10 demonstrates that using the extracted GradCams as a training phase predictor is competitive, with an *r*-squared value achieving its maximum value of $r2 = 0.5$ in the second run. This changing behavior in successfully completing the MI tasks over the runs has already been observed, as suggested in [64].

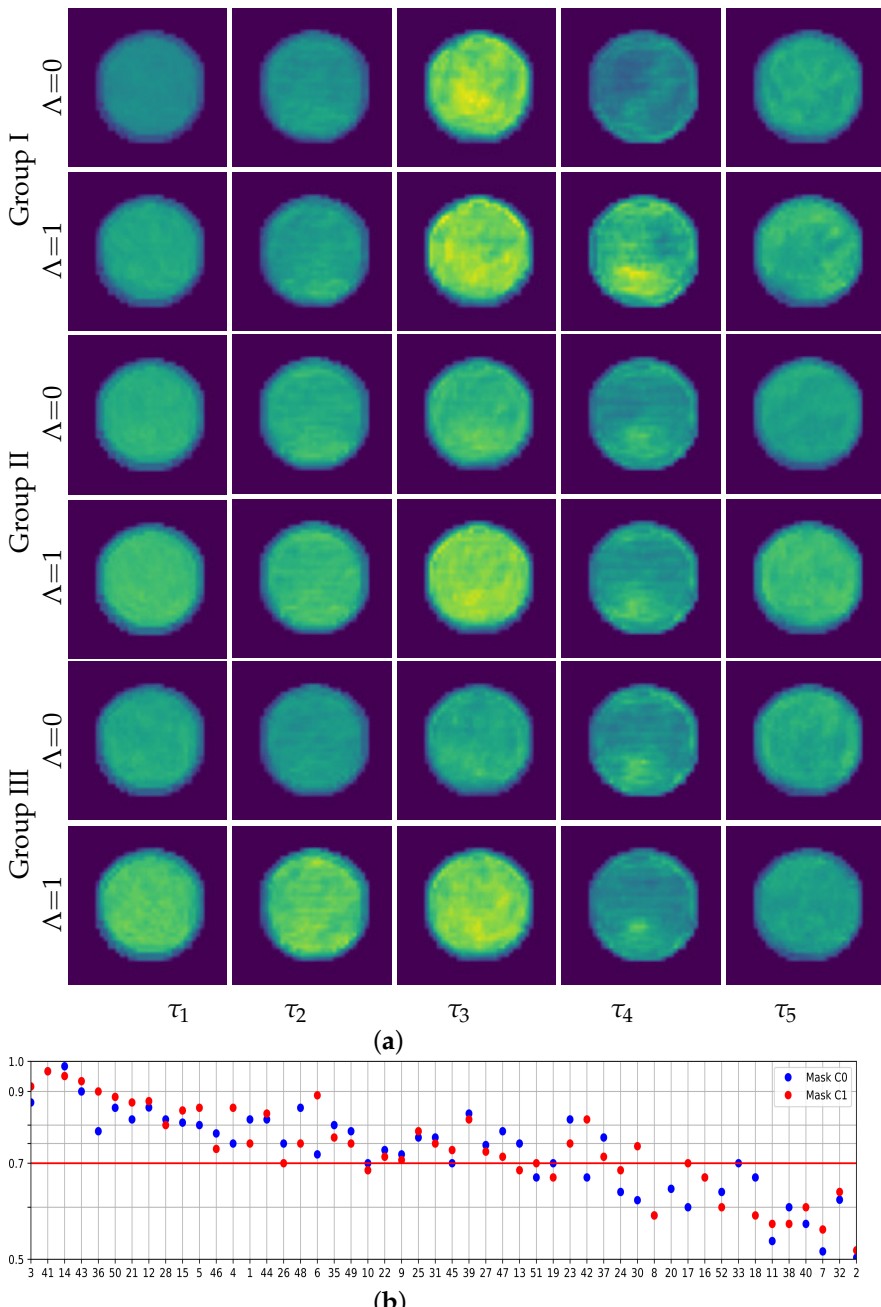

**Figure 9.** Contribution of GradCam++ maps performed by the skills-ranked groups for either label ($\Lambda = 0$—left hand, $\Lambda = 1$—right hand) along the timeline sequence. The Mask C0 and C1 display the achieved accuracy using GradCam++ representation of left and right-hand label as filter mask, respectively. (**a**) Timeline of GradCam++ maps; (**b**) hemisphere-based accuracy of D&W CNN classifier.

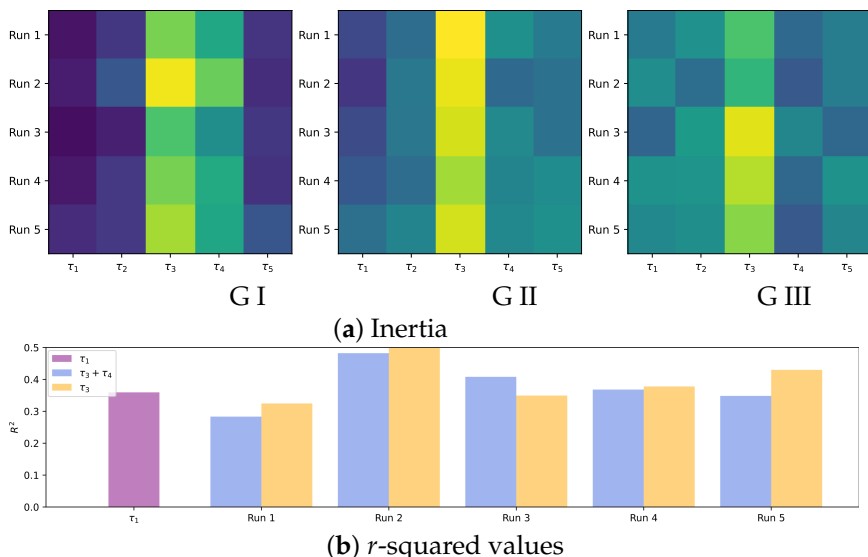

(**a**) Inertia

(**b**) *r*-squared values

**Figure 10.** Prediction ability of extracted GradCam++. (**a**) Inertia estimation over training runs. (**b**) Correlation between the features extracted from the MI onset responses.

## 5. Discussion and Concluding Remarks

This study explores the ability to distinguish between motor imagery tasks and the interpretability of the brain's ability to produce elicited mental responses with improved accuracy. To this end, we develop a Deep and Wide convolutional neuronal network fed by a set of topoplots derived from the multi-channel EEG data for two brain rhythms ($\mu$, $\beta$) through the feature extraction methods (WT and FBCSP). Then, a visualization approach based on gradient-based activation maps is employed to interpret the learned neural network weights discriminating between MI responses. In order to take into account the intra-subject variability of neural responses over time, each visualization map is extracted at different intervals along the MI paradigm timeline. Additionally, we cluster the common dynamic spatial representations of the extracted GradCam maps across the subject set, together with the accuracy values of the D&W NN classification, to account for the poor MI-BCI skills phenomenon. The results obtained from the evaluated GigaScience Database of motor-evoked potentials show the potential of the developed approach to improve the meaningful physiological explanation of motor imagery skills in aspects like neural synchronization between rhythms, brain lateralization, and the prediction ability of extracted GradCam maps.

After the evaluation stage, however, the following points are worth mentioning:

*Preprocessing and Extraction Topoplots*: Preprocessing of EEG signals is commonly implemented by three steps: artifact removal, channel selection, and frequency filtering. Based on the fact that DL can extract useful features from raw and unfiltered data, as a rule, the first two steps are not performed [65,66]. As channel selection may enable lower generalization errors in DL [67], we compute the topoplot representation from the whole set of EEG channels to visualize the spectral power variations on the scalp averaged over each MI interval. The topoplots extracted by FBCSP and CWT algorithms are inputs to a Convolutional Neural Network with a Deep and Wide architecture that is more generic for decoding EEG signals, delivering a competitive classification accuracy[68]. The feature extraction procedures require several parameters to fix, affecting the CNN learning properties like discriminability and interpretability. However, the short-time window selected to encode the latency of brain responses must be adjusted to extract temporal EEG dynamics accurately [69]. For example, as [61] suggests, using shorter window values may increase the performance of poor-performing subjects.

*Achieved accuracy by CNN Neural-Network learner*: We take advantage of the D&W architecture for classification problems with multiple inputs so that a multi-view set of

extracted time-frequency features supports the CNN learner. The evaluated CNN classifier uses a multi-layer perceptron that includes batch normalization and fully connected layers designed to exploit the close spatial relationship present in the topoplot data set, improving performance on unseen examples. As a result, the implemented D&W CNN learner fed by the extracted feature sets significantly improves the tested GigaScience data set' accuracy, requiring a few trainable parameters. Table 2 compares the classifier performance of some recently reported works using CNN-based learners, underperforming both proposed scenarios of D&W CNN training.

**Table 2.** Comparison of bi-class accuracy achieved by state-of-the-art approaches in *GigaScience* collection. Compared approaches that include interpretability analysis are marked with ✓, else are marked with −.

| Approach | Accuracy | Interpretability |
|---|---|---|
| EEGnet [70] | 66.0 | − |
| LSTM + Optical [71] | 68.2 ± 9.0 | − |
| EGGnetv4 + EA [72] | 73.4 | ✓ |
| DC1JNN [73] | 76.50 | ✓ |
| MINE + EEGnet [74] | 76.6 ± 12.48 | ✓ |
| MSNN [75] | 81.0 ± 12.00 | ✓ |
| D&W | 76.2 ± 3.96 | ✓ |
| D&W+GC++ | **84.1 ± 3.46** | ✓ |

*Computation of GradCam++ Visualization maps*: In MI EEG, the use of GradCam sets is growing, providing insight into the electrodes' ability to discriminate the elicited tasks. Specifically, we introduce the recently proposed GradCam++ method, which enables the identification of multiple instances of the same class and multidimensional representations of inputs. In addition, the activation maps are developed for optimization of the generalized model architecture to increase the CNN's learner performance further (See the accuracy achieved by D&W CNN-M), and at the same time investigate how brain regions and neural activity are closely connected. However, there are some important pitfalls, such as the gradient vanishing problem inherent to deep architectures that tends to result in noisy activation maps and thus deteriorates the visualization ability of the process as a whole [76]. Another concern is that the CAMs with higher weights show a lower contribution to the network's output than the zero-baseline of RELU activation. This phenomenon may have two sources: the global pooling operation on the top of the gradients and the gradient vanishing problem [77]. To cope with this issue, we randomly select activation maps, up-sample them into the input size, and then record how much the target score will be if we only keep the region highlighted in the activation maps.

*Interpretability of MI Skills using GradCam++*: To improve visualization efficiency in interpreting the MI coordination skills, we validate some strategies to accommodate training spaces, such as piece-wise computation of GradCam++ maps over time, clustering MI skills using common GradCams, and splitting the labeled training set. In this way, the interpretation of the spatial contributions improves in discriminating between both labels of motor imagery recognition providing better physiological interpretability of the paradigm. Hence, the post hoc visualization method developed for CNN architectures can be coupled with neurophysiologically grounded models of the Motor Imagery paradigm.

As future work, the authors plan to fuse several CNN architectures as an alternative to the previous approach of separating temporal and spatial (like recurrent and Long Short-Term Memory Network), explicitly accounting for sequential dependencies in time. Since GradCam++ can be estimated on a trial basis, subject-specific visualization designs will be explored to address poor MI coordination skills using CNN frameworks with transfer learning [78]. In addition, we plan to investigate the combination of the CNN-based feature extraction approach and fuzzy classifiers due to the great versatility and transversality of

fuzzy techniques, which, as is well known, are data independent [79]. In particular, we will explore the novel fuzzy classification based on fuzzy similarity techniques demonstrating good performance on the assessment of the mechanical the integrity of a steel plate [80].

**Author Contributions:** Conceptualization, D.F.C.-H., A.M.Á.-M. and G.C.-D.; methodology, D.F.C.-H., A.M.Á.-M. and G.C.-D.; validation, D.F.C.-H. and A.M.Á.-M.; data curation, D.F.C.-H.; writing—original draft preparation, D.F.C.-H. and G.C.-D.; writing—review and editing, D.F.C.-H., A.M.Á.-M. and G.C.-D. All authors have read and agreed to the published version of the manuscript.

**Funding:** This research manuscript is developed supported by "Convocatoria Doctorados Nacionales COLCIENCIAS 785 de 2017" (Minciencias). Also, under grants provided by the Minciencias project: Herramienta de apoyo a la predicción de los efectos de anestesicos locales via neuroaxial epidural a partir de termografía por infrarrojo code 111984468021.

**Institutional Review Board Statement:** Not applicable.

**Informed Consent Statement:** No applicable since this study uses duly anonymized public databases.

**Data Availability Statement:** The databases used in this study are public and can be found at the following links: *GigaScience*: http://gigadb.org/dataset/100295, accessed on 1 August 2021.

**Conflicts of Interest:** The authors declare no conflict of interest.

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
