# Peer review of "Image-Based Learning Using Gradient Class Activation Maps for Enhanced Physiological Interpretability of Motor Imagery Skills"

_applsci, doi:10.3390/app12031695_

Round 1

Reviewer 1 Report

*) All mathematical formulas should be numbered.

*) Each non-original mathematical formula should be associated with a relevant bibliographic reference.

*) There are some typos in the text of the paper. Please delete them.

*) The Euclidean norm was used in the formula on line 199. What suggested to the authors to use Euclidean space as a functional space? Was a study done that revealed the optimal functional space? Please specify.

*) (6) how was it obtained? Is it already known in the literature? Or is it a formulation proposed by the authors?

*) Classification of MI Tasks is certainly a problem of great interest and the approach used seems effective and efficient. However, great care must be taken when the available data are affected by uncertainties and / or inaccuracies. In these cases, fuzzy classifiers should be used. Without prejudice to the fact that such an application goes beyond the work done by the authors, I ask the authors to put in the text a sentence that highlights this important possibility by inserting the following relevant works in the bibliography:

doi: 10.1016/j.renene.2018.05.008

doi: 10.1515/phys-2020-0159 

These two works are interesting because they offer the latest generation of fuzzy classifiers. Even if applied to problems different from the one proposed by the authors, they underline the great versatility and transversality of fuzzy techniques which, as is well known, are data independent. Furthermore, the second recommended paper concerns a new concept of fuzzy classification based on fuzzy similarity techniques, currently a novelty in the panorama of international scientific research.

Reviewer 2 Report

The paper is very good, it is an interesting approach.  However, I found the following issues:

First using images, 2D images, of topoplots to analyze EEG signals (1D signals) is an interesting approach in itself.  Are there any other similar methods ?  Are there other methods using plots of EEG or scalograms or spectrograms of signals to perform the analysis of EEG ?  Are there any other manuscript using waveforms of EEG to classify or identify them in BCI contexts?  I couldn't find any comment on the introduction, nor on the provided references.

It is clear that this is a continuation of previous work, but the manuscript relies heavily on previous works to explain important aspects of this manuscript.  For instance, more detailed information should perhaps be provided to describe the GradCam++ approach.  This would help the manuscript more to be self-contained.

BCI-illiteracy: I do not agree with how the authors handle BCI-illiteracy.  First the concept itself is currently being debated as such (https://link.springer.com/article/10.1007/s11948-018-0061-1). Second, authors use this term to describe the natural variability, the inter-subject variability of BCI systems due to differences in the generated signals.  Hence, adaptive classifiers are required to be able to perform well for every subject.  My suggestion will be to throw the term away, and highlight the benefits of the adaptive approach of the proposed method to deal with Inter-Subject variability.

Definitions of section 2.1:  The mathematical notation is confusing.  Several variables are used without being defined (lambda eq on 114).  All the notation seems very confusing.  I believe authors should clarify better:  how are you Preprocessing the EEG signals ? How do you create the input topomaps that you use to feed the CNN?  What kind of information are you getting out of the CNN (classification outputs) ?  How do you divide the dataset to perform training and testing?  Are there any connection with the clinical practice of analyzing  EEG signals ?

Some issues found along the manuscript:

Line 31: What do authors mean by "high-quality mental imagery".  I think a proper term will be mental imagery signals with a higher SNR.

Line 68: The issue of explainability may require a better explanation in terms of the implications that it may have.   For instance, why is that important ?  Only because the  intelligible property required by any NN?

Line 112: Are you feeding the topomaps into a CNN layer and use that to discriminate between MI signals ?   If that is the case, I believe authors should specify more clearly how they create the topomaps, what is their size.  Sometimes it is good to add a figure with the structure of the network, the general layout where input-output relationships are clarified. 

Line 236: "though only 60 ARE available"

Line 346: Figure 6: is this a topomap representing activations on different regions on the scalp ?

Round 2

Reviewer 1 Report

All suggestions have been implemented. So, in my opinion, the paper deserves publication.

Reviewer 2 Report

Dear Authors, 

The modification on the manuscript have been widespread and satisfactory.  However there are two points that are still very confusing, at least from my point of view.

Line 41: Authors insist on the usage of BCI-illiteracy concept.  I would suggest try to review the idea or at least provide some explanation about how the concept, or at least one definition, leads to some inconsistency within the manuscript.  Let me put it forward: in a sense, being "illiterate" means that a person is unable to read.  Hence, if we want to improve the "performance" of a person that is illiterate we cannot generate bigger letters, we need to teach, to train the subject in order to avoid to overcome the illiteracy.  It is something that must be done on the person-side of a BCI system.  Using the same approach, there is no way a BCI system can improve BCI-illiteracy through better classifiers, because the concept itself is pointing towards something that a person may lack (and there is no way a classifier can improve).  This is the reason why the whole concept may be wrong.

Line 100: Authors state "This study explores the ability to distinguish between motor imagery tasks and the interpretability of the brain’s ability to produce elicited mental responses with improved accuracy. As a continuation of the work in [47], we develop a Deep&Wide Convolutional Neuronal Network (D&W CNN) fed by a set of image-based representations (topoplots) extracted by the methods of wavelet transform and FBCSP from the multichannel EEG data within the brain rhythms.".  I would like to ask Authors how in this proposal do you decode the mental responses produced by the brain and how do Authors know that the improved accuracy is a reliable mapping of the brain ability (and not something else which originates from the interplay of the methods proposed here) ?
